# NOVELTY DETECTION WITH GAN

## ABSTRACT

The ability of a classifier to recognize unknown inputs is important for many classification-based systems. We discuss the problem of simultaneous classification and novelty detection, i.e. determining whether an input is from the known set of classes and from which specific class, or from an unknown domain and does not belong to any of the known classes. We propose a method based on the Generative Adversarial Networks (GAN) framework. We show that a multi-class discriminator trained with a generator that generates samples from a mixture of nominal and novel data distributions is the optimal novelty detector. We approximate that generator with a *mixture generator* trained with the Feature Matching loss and empirically show that the proposed method outperforms conventional methods for novelty detection. Our findings demonstrate a simple, yet powerful new application of the GAN framework for the task of novelty detection.

## 1 INTRODUCTION

In recent years we have witnessed incredible progress in AI, largely due to the success of Deep Learning, and more specifically Supervised Deep Learning. One of the basic requirements from a good supervised learning algorithm is generalization - the ability to classify input that is reasonably similar to the training data. However, usually there are no requirements whatsoever on how the classifier should behave for new types of input that differ substantially from the data that are available during training. In fact for such *novel* input the algorithm will produce erroneous output and classify it as one of the classes that were available to it during training. Ideally, we would like that the classifier, in addition to its generalization ability, be able to detect novel inputs, or in other words, we would like the classifier to say, *"I don't know."*

*Novelty detection* can be defined as the task of recognizing that test data differs in some manner from the data that was available during training. The problem of novelty detection arises in many fields and is closely related, but not identical, to the problems of *anomaly detection* (also *outlier detection*), where the goal is to recognize anomalous examples in a dataset Chandola et al. (2009). Novelty detection is a hard problem that has received relatively little attention in the ML literature. Nevertheless, in our opinion novelty detection should be a central part of every recognition system.

For a comprehensive in-depth review of the topic of novelty detection, we refer the reader to Pimentel et al. (2014). Popular conventional novelty detection methods such as *Probabilistic methods* Pimentel et al. (2014), which perform density estimation of the examples from a nominal class, or *Domain-based methods* Schölkopf et al. (2001); Tax & Duin (2004), which re-formulate novelty detection as a "one-class classification" problem and define a boundary around the nominal class, do not scale well to large high-dimensional datasets. Another class of novelty detection methods is *distance-based methods*, which assume that the nominal data are tightly clustered, while novel data occur far from their nearest neighbors. These methods require computationally expensive clustering or nearest neighbors search. One of the major drawbacks of conventional novelty detection methods is that at test time, they are separated from the classification algorithm and therefore increase the overall computational and design complexity of the system. Furthermore, combining novelty detection with a multi-class classifier into a single algorithm may leverage important intra and inter-class information in the training data which can benefit both tasks.

In this paper we are interested in methods for simultaneous classification and novelty detection. A popular heuristic approach to simultaneous classification and novelty detection is thresholding the maximum of estimated class probability Hendrycks & Gimpel (2017), or alternatively, to threshold the

entropy of the estimated probability distribution. Another practical approach is to collect "background-class" samples, that is, samples that are not from the nominal set and hopefully represent novel data. In this case, novelty detection can be reduced to a supervised learning problem. Unfortunately, this solution requires collecting a large set of background inputs - a time-consuming task and moreover, it is very difficult to sample a large enough background class that will represent all possible novel examples.

We embrace the "background-class" sampling idea and ask *is it possible to generate novel data*? To the best of our knowledge, no method for novelty detection is based on *generating* novel examples. In this paper we examine whether *Generative Adversarial Networks* (GAN), a popular generative framework that can be used to generate novel examples. The GAN framework was proposed by Goodfellow et al. (2014), as a generative modeling method, mostly used for generating realistic samples of natural images. More specifically, GAN is an approach to generative modeling where two models are trained simultaneously: a generator and a discriminator. The task of the discriminator is to classify an input as either the output of the generator ("fake" data), or actual samples from the underlying data distribution ("real" data). The goal of the generator is to produce outputs that are classified by the discriminator as "real", or as coming from the underlying data distribution.

In some formulations of GANs Odena (2016); Salimans et al. (2016); Springenberg (2015), the discriminator is trained to classify data not only into two classes "real" and "fake", but rather into multiple classes. If the "real" data consists of $K$ classes, then the output of the discriminator is $K + 1$ class probabilities where $K$ probabilities corresponds to $K$ known classes, and the $K + 1$ probability correspond to the "fake" class.

In this paper, we propose to use a multi-class GAN framework for simultaneous classification and novelty detection. If during training the generator generates *a mixture of nominal data and novel data*, the multi-class discriminator learns to discriminate novel data from nominal data and essentially became a novelty detector. At test time, when the discriminator classifies *a real example* to the $K + 1^{th}$ class, i.e., class which represented "fake examples" during training, this the example is most likely a *novel example* and not from one of the $K$ nominal classes. In fact, we prove that in this case the discriminator become an *optimal novelty detector* (for a given false positive rate). We approximate such a *mixture generator* with a generator trained with specifically designed loss functions. In Section 2 we provide background and a theoretical justification to our proposed method and its connection to existing novelty detection methods. We validate the proposed method by a set of experiments in Section 3.

## 2 Novelty detection with GANs

### 2.1 Novelty detection

In the novelty detection task, we assume that during inference, the classifier may be tested on data samples, which do not belong to the nominal data distribution $p_{data}(x)$, or in other words, novel data from $p_{novel}(x)$ and not from one of the $K$ classes that the classifier is supposed to classify. If we had access to both $p_{data}(x)$ and $p_{novel}(x)$, the densities of the nominal data and novel data, then according to Neyman-Pearson's Lemma (Lehmann & Romano, 2005), the optimal novelty detection test for a given false positive rate $\alpha$ can be achieved by thresholding the likelihood ratio $\frac{p_{data}(x)}{p_{novel}(x)}$ at an appropriate value. In practice we don't know both distributions. A "second best" solution would have to have access to both nominal and novel examples, then novelty detection can be reduced to a supervised binary classification problem. However, in practice we do not have access to novel examples.

One of the standard approaches to novelty detection is to estimate a level set of the nominal density $p_{data}(x) > \alpha$, and to declare test points outside of the estimated level set as novel. Density level set estimation is equivalent to assuming that novelties are uniformly distributed on the support of $p_{data}(x)$. Unfortunately, these methods are difficult to implement since they require estimating the high-dimensional density of the nominal data. Level set methods are closely related to one-class classification methods Schölkopf et al. (2001); Tax & Duin (2004) and in general can be reduced to binary classification problem between nominal data and artificially generated sample from uniform distribution on the data domain, as was discussed by Steinwart et al. (2005).

Blanchard et al. (2010) discussed a *Semi-Supervised Novelty Detection* (SSND) problem, where in addition to the training dataset of nominal examples we have access to an *unlabeled dataset that contains a mixture of nominal and novel examples*. The authors showed that in this case it is possible to obtain an optimal (for a given false positive rate) novelty detector. Indeed, if we have access to nominal data distribution $p_{data}(x)$ and a mixture of the form $\pi p_{novel}(x) + (1 - \pi)p_{data}(x)$ then

$$\frac{\pi p_{novel}(x) + (1 - \pi)p_{data}(x)}{p_{data}(x)} = \frac{\pi p_{novel}(x)}{p_{data}(x)} + (1 - \pi) \tag{1}$$

which leads to an appropriately scaled optimal novelty detector. Therefore, in the SSND case, novelty detection can still be reduced to a supervised classification problem. This is in contrast to the density estimation problem in which no novel data is available. Of course the problem remains to estimate the likelihood ratio from the training data.

While semi-supervised novelty detection provides interesting insights, how can these ideas be applied to the fully unsupervised novelty detection problem where only nominal data are available during training? Level set methods can be solved by generating artificial examples from uniform distribution on data domain and formulating binary classification problem. But, *is it possible to generate artificial examples from some non-uniform distribution which better represent novel data than uniform distribution*? and *Is it possible to generate examples from the mixture of nominal and non-uniform distributions to apply the SSND ideas*? And finally, *is it possible to combine these ideas for novelty detection with multi-class classification*? In the next sections we demonstrate how we can leverage the GAN framework to generate examples from the mixture of nominal and non-uniform distributions and use the GAN's discriminator as a simultaneous multi-class classifier and novelty detector.

## 2.2 GENERATIVE ADVERSARIAL NETWORKS (GANs)

Generative Adversarial Networks Goodfellow et al. (2014) is a recently proposed approach for generative modeling. The main idea behind GANs is to have two competing differentiable functions, usually implemented as neural network models. One model, which is called the generator $G(z; \theta^G)$, maps a noise sample $z$ sampled from some prior distribution $p(z)$ to the "fake" sample $x = G(z; \theta^G)$; $x$ should be similar to a "real" sample sampled from the nominal data distribution $p_{data}(x)$. The objective of the other model called the discriminator $D(x; \theta^D)$ is to correctly distinguish generated samples from the training data which samples $p_{data}(x)$. This is a minimax game between the two models with a solution at the Nash equilibrium. For an excellent overview on GANs, see Goodfellow (2016). Unfortunately, there is no closed-form solution for such problems. Therefore, the solution is approximated using an iterative gradient-based optimization of the generator and the discriminator functions. The discriminator is optimized by maximizing:

$$\max_{\theta^D} \mathbb{E}_{x \sim p_{data}(x)}[\log D(x; \theta^D)] + \mathbb{E}_{z \sim p(z)}[\log(1 - D(G(z; \theta^G); \theta^D))] \tag{2}$$

and the generator is optimized by minimizing:

$$\min_{\theta^G} \mathbb{E}_{z \sim p(z)}[\log(1 - D(G(z; \theta^G); \theta^D))] \tag{3}$$

For a fixed generator $G(z)$ we can analytically derive that the optimal discriminator will take the form:

$$D_G^*(x) = \frac{p_{data}(x)}{p_{data}(x) + p_g(x)} \tag{4}$$

Since the introduction of Generative Adversarial Networks by Goodfellow et al. (2014), multiple variants of GANs were published in the literature. GANs were applied to various interesting tasks such as realistic image generation Radford et al. (2015), text-to-image generation Reed et al. (2016), video generation Vondrick et al. (2016), image-to-image generation Isola et al. (2016), image inpainting Pathak et al. (2016), super-resolution Ledig et al. (2017), and many more. In almost all of these applications, only the generator $G$ is used at a test time, while the discriminator $D$ is trained for the sake of training the generator and is discarded at test time.

In contrast, Salimans et al. (2016) introduced a GAN based semi-supervised classifier *(SSL-GAN)*. They showed that in case where only a small fraction of the real examples have labels, and the bulk

of the real data is unlabeled, their GAN framework is able to train a powerful multi-class classifier, which is the discriminator $D$. In order to improve the GAN convergence, (Salimans et al., 2016) proposed to optimize the generator by minimizing a *Feature-Matching* loss:

$$L_{FM}(x) = \min_{\theta^G} ||\mathbb{E}_{x \sim p_{data}(x)}[\mathbf{f}(x)] - \mathbb{E}_{z \sim p(z)}[\mathbf{f}(G(z; \theta^G))]|| \qquad (5)$$

where $\mathbf{f}(x)$ is an intermediate layer of the fixed discriminator used as a feature representation of $x$. Optimizing a generator using the Feature Matching loss results in samples which are not of the best visual quality, but the resulting multi-class discriminator performs well for supervised classification and for semi-supervised classification as well.

### 2.3    MIXTURE GENERATOR FOR NOVELTY DETECTION

In the semi-supervised settings of multi-class classification, Salimans et al. (2016) showed that SSL-GAN trained with the Feature Matching loss (eq.5) improves the classification accuracy of the discriminator. This improvement occurs even though the generated samples are not of the best visual quality. Dai et al. (2017) addressing the question "why does the Feature Matching loss improves the semi-supervised discriminator" made two important observations: In Proposition 1 Dai et al. (2017) show that when the generator is *perfect*, i.e. $p_g(x) = p_{data}(x)$, it does not improve the generalization performance of the SSL-GAN discriminator over the supervised learning training without GAN. In Proposition 2 Dai et al. (2017) defined the *complement generator*, a generator which generates samples with feature representation $\mathbf{f}(x)$ distributed in a complementary region to the distribution support of the features of the real data. They show that under mild assumptions when training a multi-class discriminator with the complement generator the discriminator places the real-classes boundaries in low-density areas of the features distributions of the real data. The conclusion is that only a *bad generator*, where $p_g(x) \neq p_{data}(x)$ and which generates samples outside of the high-density areas of the real data distribution, is able to improve semi-supervised learning as was demonstrated by Salimans et al. (2016).

The results of Salimans et al. (2016) and Dai et al. (2017) led us to propose the following definition of a *mixture generator*:

**Definition 1.** *Mixture generator is a generator with a mixture distribution $p_g(x) = \pi p_{other}(x) + (1 - \pi)p_{data}(x)$ of the true data distribution $p_{data}(x)$ and some other distribution $p_{other}(x)$, such that there exists a non-empty region $\Omega$ where $\{\forall x \in \Omega, p_{other}(x) > p_{data}(x), p_{data}(x) \leq \epsilon\}$, for some $\epsilon > 0$.*

In other words the mixture generator is a generator that generates a mixture of true data distribution $p_{data}(x)$ and some other data $p_{other}(x)$, where at least part of the $p_{other}(x)$ probability mass is concentrated in lower-density regions of $p_{data}(x)$.

In general, any generator distribution $p_g(x)$ that is different from $p_{data}(x)$, can be represented as a *unique mixture* of $p_{data}(x)$ and some other distribution, which by itself cannot be represented as a (nontrivial) mixture of $p_{data}(x)$ with another distribution (see Proposition 5 by Blanchard et al. (2010)). However not any generator with $p_g(x) \neq p_{data}(x)$ is a mixture generator. The requirement is that at least part of the $p_{other}(x)$ probability mass is concentrated in lower-density regions of $p_{data}(x)$ excludes cases where $p_g(x) \neq p_{data}(x)$ but generates samples in high-density regions of $p_{data}(x)$ as happens in the case of mode collapse.

The mixture generator can be viewed as a relaxed version of the theoretical complement generator defined by Dai et al. (2017): every complement generator is a mixture generator with $\pi = 1$ and where $p_{other}(x)$ and $p_{data}(x)$ have disjoint support, but the opposite is not true. We can also consider the *degenerate mixture generator*: in the degenerate case $p_{other}(x)$ is a uniform distribution over the real data domain and $\pi = 1$ (assuming $p_{data}(x)$ is not uniform by itself). In this case no learning for such generator is required, and the generator only need to be able to generate uniformly distributed samples in data domain.

For novelty detection we would like the generator to generate samples from the (unknown) distribution of the novel data $p_{novel}(x)$. This allows us to solve the novelty detection problem using the following observation:

**Proposition 1.** *For a fixed mixture generator $G$, if $p_{other}(x)$ defines a distribution of the novel data, i.e. $p_{other}(x) = p_{novel}(x)$, then the optimal discriminator $D_G^*(x)$ is also an optimal (for a given false positive rate) novelty detector.*

The proof of Proposition 1 follows trivially from the definition of the mixture generator, eq. 4 of the optimal discriminator for a fixed generator, and eq. 1 of the optimal novelty detector as in the SSND case. Indeed, if $p_g(x) = \pi p_{novel}(x) + (1 - \pi)p_{data}(x)$ then we can invert eq. 4 and we have

$$\frac{1 - D_G^*(x)}{D_G^*(x)} = \frac{p_g(x)}{p_{data}(x)} = \frac{\pi p_{novel}(x) + (1 - \pi)p_{data}(x)}{p_{data}(x)} = \pi \frac{p_{novel}(x)}{p_{data}(x)} + (1 - \pi) \quad (6)$$

which as in eq. 1 is an optimal novelty detector for a given false positive rate.

The question is which mixture generator is capable of generating data from the unknown distribution $p_{novel}(x)$ of the real novel data. Unfortunately, without any knowledge of the specific distribution and without labeled real examples from this distribution, or at least unlabeled examples from a mixture of real novel data and nominal data, it is impossible to design a generator which will generate samples from it[1]. Nevertheless, we can train a mixture generator with specific properties of $p_{other}(x)$ which will help novelty detection.

In the case of the degenerate mixture generator, i.e. when the generator generates samples uniformly distributed over data domain, the discriminator in eq. 6 is a level set novelty detector. As we mentioned in section 2.1 such reduction of the level-set novelty detection to a classification problem is well known in the literature (Steinwart et al. (2005) and references therein). However, it is clear that sampling novel examples from a uniform distribution in a very high dimensional space is not an efficient strategy for training novelty detector.

Therefore, we want a mixture generator that will generate novel data distributed in nearby surrounding or at low-density regions of the true data manifold. To train such a mixture generator we need a loss function that encourage the generator to do that. For example Dai et al. (2017) proposed a loss function based on the Kullback-Leibler (KL) divergence between the distributions of the generator distribution and distribution of the data. More specifically they propose to minimize:

$$\min_{\theta^G} -\mathcal{H}\left(p_g(x; \theta^G)\right) + \mathbb{E}_{x \sim p_g(x; \theta^G)} \log p_{data}(x)\mathbb{I}[p_{data}(x) > \epsilon] + L_{FM}(x; \theta^G) \quad (7)$$

where $\mathcal{H}(\cdot)$ is the entropy function and $\mathbb{I}[\cdot]$ is the indicator function. This loss function is designed to produce a generator with a distribution which on one hand has support which does not intersect with high density regions of the real data (second term), but still close to the data manifold (Feature Matching loss as a third term). Unfortunately, to train a generator with the loss function in eq. 7, we need to estimate $p_{data}(x)$, the same problem which we wanted to avoid in conventional novelty detection methods.

In their paper Dai et al. (2017) experimentally demonstrated on two synthetic datasets that a generator that is trained with a Feature Matching loss eq. 5 is able to generates both samples that fall onto the data manifold, and samples which are scattered in the nearby surrounding of the data manifold, i.e. a mixture of real data and a data outside of data manifold. The empirical results of Dai et al. (2017) suggest that a generator that is trained with the Feature Matching loss only, is by itself a mixture generator. Moreover, we know from the results of Dai et al. (2017) that when the generator generates samples that are scattered around and in the low-density areas of the data manifold, the discriminator classification boundaries becomes tighter, thus improving the discriminator's ability to detect real novelties. In Section 3 we empirically demonstrate that when a multi-class discriminator trained with a mixture generator that was trained with the Feature Matching loss, has an impressive ability to detect real novel examples.

To summarize, we propose to train a GAN in the multi-class setting with the Feature Matching loss or some other similar loss which facilitates the generator to be a mixture generator as in Definition 1 to solve the problem of simultaneous classification and novelty detection. When the multi-class discriminator of the network classifies an example as "fake", it means that most likely this example is a novel example, otherwise the class with the highest probability is the classification result. The novelty detection scores are the "fake" class probability or a related quantity such as $\frac{D_G(x)}{1 - D_G(x)}$, the

---

[1]Cherti et al. (2016) empirically demonstrated that a GAN that was trained with a regular loss function i able to generate out-of-class realistic images, e.g., a GAN trained on MNIST digits generated Latin letters

ratio between "real" and "fake" class-probabilities. The novelty detection scores are thresholded to achieve a desired false positive rate. The GAN based training produce a unified multi-class classifier and novelty detector with minimal additional computation overhead at inference time. Moreover, using unlabeled samples as in Salimans et al. (2016) improve both the multi-class classifier and the novelty detector.

## 3 EXPERIMENTS

We perform an experimental evaluation of the proposed GAN-based novelty detection method and compare it to several methods for simultaneous classification and novelty detection on the MNIST and CIFAR10 datasets. As a comparison metric we use the Area Under the Receiver Operating Characteristic curve (AUROC) of novelty detection scores.

The first method that we compare to is based on the estimated class probabilities of a multi-class classifier. Hendrycks & Gimpel (2017) suggested to use the maximum of the estimated (softmax) probability as a baseline for out-of-distribution data (i.e. novelty) detection score. Another popular novelty score is the entropy of the estimated probabilities, which takes into account all the predicted class probabilities. These scores are highly correlated.

In addition, we compare to methods that rely on nearest-neighbors distance analysis in feature space derived from the trained multi-class classifier (e.g. last convolutional layer of CNN). We use normalized k-NN distance as a novelty score as described by Ding et al. (2014). The kNN novelty score of a test data point $x$ is a distance to the $k^{th}$ nearest neighbor in the nominal training dataset $NN_k(x)$, where $NN_k()$ denotes the $k$ nearest neighbors, normalized by k-NN distance between the $NN_k(x)$ to its $k$ nearest neighbors $NN_k(NN_k(x))$:

$$\frac{d(x, NN_k(x))}{d(NN_k(x), NN_k(NN_k(x)))}. \tag{8}$$

For $k = 1$, $d$ is an $l_2$ distance in a feature space. For $k > 1$, $d$ denotes the average distance to the $k$ nearest neighbors. We evaluate this method using $k = \{1, 5\}$.

Additionally we experimented with the OCSVM and the SVDD methods Schölkopf et al. (2001); Tax & Duin (2004) implemented in LIBSVM library Chang & Lin (2011). We tried various kernels and parameters values, but failed to achieve competitive results, even relative to the kNN methods. Our findings are supported by Ding et al. (2014) where kNN methods outperformed domain-based methods on multiple datasets. The fact that domain-based method struggling in applications involving high-dimensional spaces was also noted by Pimentel et al. (2014).

For the GAN based novelty detection score which we call *ND-GAN*, we trained the SSL-GAN model as described by Salimans et al. (2016) using modified publicly available implementation of SSL-GAN[2] and using Theano Theano Development Team (2016). The mixture generator was trained with the Feature-Matching loss eq.5, and the novelty score of the ND-GAN was defined as the ratio: $\frac{D_G(x)}{1-D_G(x)}$, the ratio between "real" class and "fake" class probabilities. To make a fair comparison, when comparing to the competing methods, we trained the multi-class classifiers with the same architectures as the discriminator.

### 3.1 MNIST

We performed two experiments with MNIST dataset, novelty detection from other datasets and a holdout novelty detection test. First, we identify novelties with respect to the MNIST datasets by running MNIST trained classifiers on the Omniglot dataset of images of handwritten characters from Lake et al. (2015), and notMNIST dataset images typeface characters from Bulatov (2011). 10000 MNIST test images are used as positive examples, while images from Omniglot and notMNIST datasets are used as a negative examples. Finally, as a controlled experiment we use the ICDAR 2003 Lucas et al. (2003) dataset of handwritten Latin letters and digits. The ICDAR 2003 digits which are different from MNIST, serve as an independent validation dataset. For the GAN-based novelty detector we follow the SSL-GAN framework and use only 100 labeled examples from each class. Both

---

[2]https://github.com/openai/improved-gan

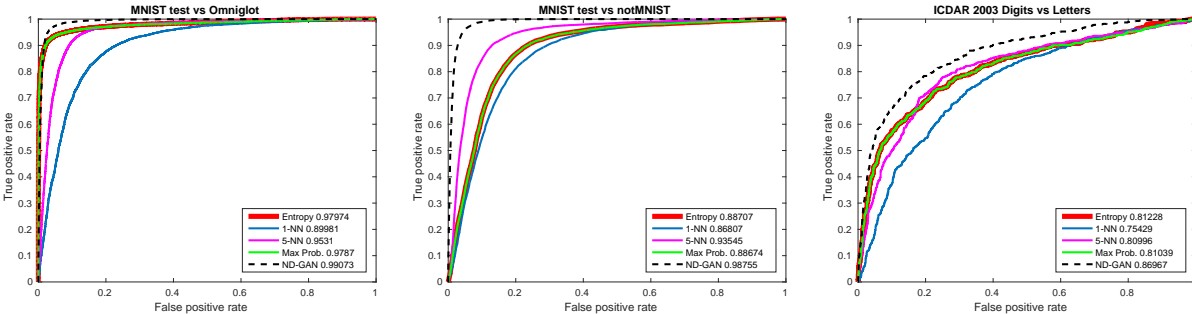

Figure 1: From left to right, ROC curves and AUROC numbers, of training a model on MNIST digits and testing for novelty against the Omniglot, notMNIST. In the ICDAR-2003 dataset, digits were used as nominal examples while letters were used as novel examples.

the generator and the discriminator have 5 fully connected hidden layers each. Weight normalization Salimans & Kingma (2016) was used and Gaussian noise was added to the output of each layer of the discriminator. To evaluate the other methods (kNN, entropy and max-probability), we trained a standard supervised network with the same architecture. The k-NN distances were computed from the 250 dimensional feature vectors of the last fully connected layer. For the ICDAR 2003 datasets the following ambiguous letters {o, O, i, I, z, Z, S, s,l} were excluded from the experiment. The ROC curves of the experiments are depicted in Figure 1. We see that the ND-GAN novelty score outperforms the other novelty detection methods in all experiments.

In the holdout experiment, we train an SSL-GAN model and multi-class classifier with the same architecture as discriminator for each of the ten digits. Every model is trained on nine out of the ten classes. During testing, we compute the novelty scores for all of the classes, including the holdout class which is considered to be novel (negative). For the holdout class testing is performed on both the train and test examples in order to balance the number of nominal and novel examples. In Table 1 we present the AUROC of ND-GAN and the other competing methods. We see that in seven out of ten holdout experiments, ND-GAN outperforms the other methods, as well as the mean result.

Table 1: MNIST holdout experiment: comparison of different novelty detection methods on the MNIST dataset. In each of the ten tests, the model was trained with the standard MNIST training set, excluding one hold-out digit. The AUROC is computed from a balanced test-set in which the nine digits are considered nominal data and the hold-out digit is novel.

| HoldOut | 0 | 1 | 2 | 3 | 4 | 5 | 6 | 7 | 8 | 9 | mean |
|---|---|---|---|---|---|---|---|---|---|---|---|
| Entropy | 0.966 | **0.984** | 0.965 | 0.961 | **0.947** | 0.957 | **0.955** | 0.970 | 0.974 | 0.958 | 0.964 |
| max prob. | 0.965 | 0.984 | 0.964 | 0.961 | 0.946 | 0.957 | 0.954 | 0.970 | 0.973 | 0.958 | 0.963 |
| 1-NN | 0.916 | 0.921 | 0.862 | 0.844 | 0.680 | 0.863 | 0.865 | 0.818 | 0.857 | 0.840 | 0.846 |
| 5-NN | 0.975 | 0.952 | 0.943 | 0.930 | 0.811 | 0.918 | 0.919 | 0.941 | 0.928 | 0.918 | 0.924 |
| ND-GAN | **0.992** | 0.982 | **0.966** | **0.976** | 0.936 | **0.989** | 0.947 | **0.978** | **0.976** | **0.967** | **0.971** |

## 3.2 CIFAR10 vs CIFAR100

In the CIFAR experiment we train classifiers on the CIFAR10 train datatset and test for novelties on images from the CIFAR10 test datasets (nominal examples) and the CIFAR100 dataset (novel examples) Krizhevsky (2009). The CIFAR100 dataset contains 100 fine and 20 coarse categories, which differ from the CIFAR10 categories (small overlaps in the data such as *lion-cat*, *wolf-dog*, *truck-bus*, etc. contributes equally to the error of all of the methods). For the ND-GAN novelty detector, we employ the architecture as in the SSL-GAN framework Salimans et al. (2016) with 3000 labeled examples from each class. For the discriminator a 9 layer deep convolutional network with dropout and weight normalization was used. The generator has a 4 layer deep CNN with batch normalization. For the other methods (kNN, entropy and max-probability) the same architecture as in the discriminator was used. The k-NN distances are computed from the 192 dimensional feature

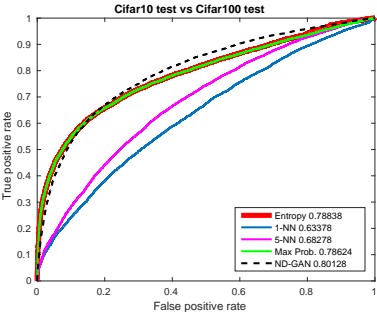

Figure 2: CIFAR10 vs CIFAR100 experiment: The CIFAR10-train-set was used to train the model. The CIFAR10 test-set was used as a nominal data and the CIFAR100 test-set as a novel data.

vectors of last convolution layer. Figure 2 depicts a ROC curves in which the CIFAR10 test set (nominal examples) is compared against the whole CIFAR100 test set (novel examples). We see that the ND-GAN novelty score performs comparably to other methods. We also compare each of the 20 coarse categories in CIFAR100 separately. The results of this experiment are presented in Table 2. We see that in 13 out of 20 coarse CIFAR100 categories ND-GAN outperforms the other methods.

Table 2: CIFAR10 vs CIFAR100 experiment: comparison of different novelty detection methods on the 20 CIFAR100 coarse classes. The AUROC is computed from a balanced CIFAR10 test-set as a nominal data and train and test data of the CIFAR100 coarse classes as a novel data.

| Category | Entr. | max prob. | 1-NN | 5-NN | ND-GAN | Category | Entr. | max prob. | 1-NN | 5-NN | ND-GAN |
|---|---|---|---|---|---|---|---|---|---|---|---|
| aquatic mamls. | 0.800 | 0.798 | 0.602 | 0.639 | **0.841** | natural scenes | 0.807 | 0.803 | 0.612 | 0.662 | **0.811** |
| fish | 0.811 | 0.808 | 0.615 | 0.678 | **0.862** | herbivores | 0.780 | 0.779 | 0.586 | 0.616 | **0.783** |
| flowers | 0.788 | 0.786 | 0.676 | 0.768 | **0.793** | med. mamls. | 0.794 | 0.792 | 0.593 | 0.620 | **0.809** |
| food cont. | **0.779** | 0.777 | 0.684 | 0.758 | 0.759 | invertebrates | 0.804 | 0.801 | 0.641 | 0.682 | **0.816** |
| fruit & vegies | **0.797** | 0.794 | 0.678 | 0.756 | 0.786 | people | **0.780** | 0.779 | 0.619 | 0.650 | 0.744 |
| household dev. | 0.780 | 0.778 | 0.704 | **0.781** | 0.768 | reptiles | 0.798 | 0.795 | 0.637 | 0.687 | **0.837** |
| household fur, | 0.792 | 0.790 | 0.666 | 0.737 | **0.821** | mamls. | **0.788** | 0.787 | 0.556 | 0.556 | 0.771 |
| insects | 0.799 | 0.797 | 0.652 | 0.712 | **0.845** | trees | 0.826 | 0.822 | 0.651 | 0.725 | **0.893** |
| carnivores | **0.773** | 0.771 | 0.579 | 0.590 | 0.765 | vehicles1 | **0.741** | 0.740 | 0.614 | 0.652 | 0.738 |
| outdoor things | 0.811 | 0.807 | 0.644 | 0.699 | **0.845** | vehicles2 | 0.771 | 0.769 | 0.613 | 0.652 | **0.822** |

## 4 CONCLUSION AND FUTURE WORK

The ability to identify novelties or say "I don't know" is an important tool for many classification-based systems. In this work, we propose to solve a problem of simultaneous classification and novelty detection within the GAN framework. We propose to use a GAN with a mixture generator to turn this problem into a supervised learning problem without collecting "background-class" data.

In the case where a mixture generator generates samples from a mixture of nominal data distribution and novel data distribution, we showed that the GAN's discriminator is an optimal novelty detector. We approximate that generator with a mixture generator trained with the Feature Matching loss. This mixture generator generates samples scattered around and in the low-density areas of the data manifold, and this makes a multi-class discriminator a powerful novelty detector. We empirically validate that the performance of the proposed solution is comparable to several popular novelty detection methods, and sometimes outperforms them.

Clearly, evaluation of the proposed framework on more challenging datasets is required. As a future research direction we would like to search for new loss functions for mixture generators that will enrich the generator distribution and will improve novelty detection. Classification with asymmetric label noise problem Scott et al. (2013) is closely related to semi-supervised novelty detection, and it will be interesting to see whether the GAN framework can be used to solve this problem. Finally, it is interesting to see if the suggested framework can be applied to detecting the adversarial examples.

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
