# OpenReview forum: "Novelty Detection with GAN"
_ICLR.cc/2018/Conference — Reject_

### Official Review · AnonReviewer2 · 2017-11-23
**Promising performance, but not well-motivated, and conceptually and theoretically strange**

**Rating:** 4
**Confidence:** 4

**Review:**

This paper proposed a GAN to unify classification and novelty detection. The technical difficulty is acceptable, but there are several issues. First of all, the motivation is clearly given in the 1st paragraph of the introduction: "In fact for such novel input the algorithm will produce erroneous output and classify it as one of the classes that were available to it during training. Ideally, we would like that the classifier, in addition to its generalization ability, be able to detect novel inputs, or in other words, we would like the classifier to say, 'I don't know.'" There is a logical gap between the ability of saying 'I don't know' and the necessity of novelty detection. Moreover, there are many papers known as "learning with abstention" and/or "learning with rejection" from NIPS, ICML, COLT, etc. (some are coauthored by Dr. Peter Bartlett or Dr. Corinna Cortes), but the current paper didn't cite those that are particularly designed to let the classifier be able to say 'I don't know'. All those abstention/rejection papers have solid theoretical guarantees.

The 3rd issue is that the novelty for the novelty detection part in the proposed GAN seems quite incremental. As mentioned in the paper, there are already a few GANs, such that "If the 'real' data consists of K classes, then the output of the discriminator is K+1 class probabilities where K probabilities corresponds to K known classes, and the K+1 probability correspond to the 'fake' class." On the other hand, the idea in this paper is that "At test time, when the discriminator classifies a real example to the K+1th class, i.e., class which represented 'fake examples' during training, this the example is most likely a novel example and not from one of the K nominal classes." This is just a replacement of concepts, where the original one is the fake class in training and the new one is the novel class in test. Furthermore, the 4th issue also comes from this replacement. The proposed method assumes a very strong distributional assumption, that is, the class-conditional density of the union of all novel classes at test time is very similar to the class-conditional density of the fake class at training time, where the choice of similarity depends on the divergence measure for training GAN. This assumption is too strong for the application of novelty detection, since novel data can be whatsoever unseen during training.

This inconsistency leads to the last issue. Again mentioned in the 1st paragraph, "there are no requirements whatsoever on how the
classifier should behave for new types of input that differ substantially from the data that are available during training". This evidences that novel data can be whatsoever unseen during training (per my words). However, the ultimate goal of the generator is to fool the discriminator by generating fake data as similar to the real data as possible in all types of GANs. Therefore, it is conceptually and theoretically strange to apply GAN to novelty detection, which is the major contribution of this paper.

Last but not least, there is an issue not quite directly related to this paper. Novelty detection sounds very data mining rather than machine learning. It is fully unsupervised without a clearly-defined goal which makes it sounds like an art rather than a science. The experimental performance is promising indeed, but a lot of domain knowledge is involved in the experiment design. I am not sure they are really novelty detection tasks because the real novelty detection tasks should be fully exploratory.

BTW, there is a paper in IPMI 2017 entitled "Unsupervised Anomaly Detection with Generative Adversarial Networks to Guide Marker Discovery", which is very closely related to the current paper but the authors seem not aware of it.

---

### Official Review · AnonReviewer1 · 2017-11-27
**I like the idea in the paper, but feel the paper misses a few details and can be written more clearly**

**Rating:** 5
**Confidence:** 4

**Review:**

The paper presents a method for novelty detection based on a multi-class GAN which is trained to output images generated from a mixture of the nominal and novel distributions. The trained discriminator is used to classify images as belonging to the nominal or novel distributions. To deal with the missing data from the novel distributions the authors propose to use a proxy mixture distribution resulting from training the GAN using the Feature matching loss.

I liked the paper and the idea of using the discriminator of the GAN to detect novelty. But I do feel the paper lakes some details to better justify/explain the design choices:

1. A multi-class GAN  is used but in the formal background presentation of GANs, section 2.2 only the binary version of the discriminator is presented. I think it would be helpful if the paper is more self contained and add the discriminator objective function to complete the presentation of the GAN design actually used.
Also would be nice if the authors can comment on whether the multi-class design necessary? can't the approach presented in the paper be naively extended to a regular GAN as long as it is trained to output a mixture distribution?

2.  It is not clear to me why the Feature Matching loss results in a mixture distribution or more specifically why it results in a mixture distribution which is helpful for novelty detection? The paragraph before eq (7) and the two after hint to why this loss results in a good mixture distribution. I think this explanation would benefit from a more formal attempt of defining what is a "good" mixture distribution.

3. In addition to the above remark, generally I feel there is a gap between the definition of mixture distribution and proposition 1 to the actual implementation choice where it cannot be assumed the p_novel is known. I feel the paper would be clearer if the authors can draw a more direct connection.

4. I am missing a baseline approach comparing to a 'regular' multi-class GAN with a reject option. i.e. a GAN which was not trained to output a mixture distribution. Comparing ROC curves for the output of a discriminator from such a regular GAN to that of the ND-GAN would help to asses the importance of the discussion of the mixture distribution.

5. Is the method proposed sensitive to noise, i.e will poor quality images of known classes have a higher chance to be classified as novel classes?

Some typos such as :

'able to generates'
'loss function i able to generate'
'this the example'

---

### Official Review · AnonReviewer3 · 2017-11-27
**using GANs for detecting novel inputs; good clarity, somewhat weak comparisons.**

**Rating:** 6
**Confidence:** 4

**Review:**


The paper proposes a GAN for novelty detection (predicting novel versus nominal data), using a mixture generator with feature matching loss.  The key difference between this paper and previous is the different definition of mixture generator.  Here the authors enforce p_other to have some significant mass in the tails of p_data (Def 1), forcing the 'other' data to be on or around the true data, creating a tight boundary around the nominal data.

The paper is well written, derives cleanly from previous work, and has solid experiments.  The experiments are weak 1) in the sense that they are not compared against simple baselines like p(x) (from, say, just thresholding a vae, or using a more powerful p(x) model -- there are lots out there), 2) other than KNNs, only compared with class-prediction based novelty detection (entropy, thresholds), and 3) in my view perform consistently, but not significantly better, than simply using the entropy of the class predictions.  How would entropy improve if it was a small ensemble instead of a single classifier?

The authors may be interested in [1], a principled approach for learning a well-calibrated uncertainty estimate on predictions.    Considering how well entropy works, I would be surprised in the model in [1] does not perform even better.

pros:
- good application of GAN models
- good writing and clarity
- solid experiments and explanations

cons:
 - results weak relative to naive baseline (entropy)
 - weak comparisons
 - lack of comparison to density models


[1] Louizos, Christos, and Max Welling. "Multiplicative Normalizing Flows for Variational Bayesian Neural Networks." arXiv preprint arXiv:1703.01961 (2017).

---

### Public Comment · (anonymous) · 2017-11-26
**About experiments and bad generator**

Dear authors,

The topic of this paper is interesting, but I have following questions:

1. According to the Proposition 1, an optimal discriminator of GAN becomes an optimal novelty detector if p_g(x) = \pi p_novel(x) - (1-\pi) p_{data}(x). However, in the experiments section, the authors used SSL-GAN, which forces generator to recover the data distribution (i.e., p_g(x)=p_{data}(x)), for the GAN based novelty detection score. Therefore, it seems that SSL-GAN is not a proper choice for the experiments.

2. In case of bad generator proposed by Dai et al. (2017), it requires a density estimator p_{data}(x). If we have the density estimator p_{data}(x) such as PixelCNN, it seems that there is no need to build a novelty detector based on GAN since we can estimate a level set of nominal density using the density estimator... Do I correctly understand?

If I don't understand the paper correctly, please do not hesitate to let me know.

Thanks in advance.

---

> ### Author Response · Authors · 2017-11-28
> **Answers: experiments and bad generator (Section 2.3)**
>
> Dear anonymous,
> Thank you very much for reading our paper. Please see bellow answers to your questions.
>
> Q1: In the  experiments section we used SSL-GAN trained with Feature Matching loss. We would like to clarify our claim: we claim that a generator of SSL-GAN with Feature Matching loss is a mixture generator.  As we mentioned in Section 2.3  (page 5, paragraph 6: “In their paper Dai et al. (2017) experimentally demonstrated…”), the empirical results of Dai et al. (2017) suggest that a generator that is trained with the Feature Matching loss is a mixture generator. Moreover, as Saliman et.al (2016) show that SSL-GAN trained with the Feature Matching loss improves the classification accuracy of the multi-class discriminator. According to the Proposition 1 of Dai et al. (2017) this contradicts the  claim that a generator that is trained with Feature Matching loss converge p_{data}(x) .
>
> Q2: You are correct that if we have a density estimator then we can estimate a level set of nominal density.  Indeed, the bad generator proposed by Dai et al. (2017) requires nominal density estimation p_{data}(x). As a matter of fact we noted in Section 2.3, page 5, paragraph 5, last sentence: “Unfortunately, to train a generator with the loss function in eq. 7, we need to estimate p_{data}(x), the same problem which we wanted to avoid in conventional novelty detection methods”. We want to avoid to estimate p_{data}(x), and therefore we proposed an alternative method. As we explained in Section 2.3 we propose to use Feature Matching loss, or any other loss which produce a mixture generator without explicitly estimating nominal density.

---

> > ### Public Comment · (anonymous) · 2017-11-29
> > **Re: Answers: experiments and bad generator (Section 2.3)**
> >
> > Dear authors,
> > Thank you for your answers. However, I still have the following questions:
> >
> > Q1: As stated in the paper of Dai et al. (2017), the feature matching loss forces the generator to be close to the true distribution p_{data} not to the novel distribution p_{novel}. I think that the key elements of loss function of the bad generator which force the generator to be close to the novel distribution are entropy term and log-likelihood of density estimator (please see Section 5.3 of paper of  Dai et al. (2017)). Therefore, it is still unclear that what elements of SSL-GAN with Feature Matching loss make the generator to be close to novel distribution p_{novel}(x) since SSL-GAN also targets to recover the data distribution p_{data}...
> >
> > Q2: For the same reason in Q1, I think that the feature matching loss does not produce a mixture generator of novel distribution and true distribution.

---

### Decision · Program_Chairs · 2018-01-29
**ICLR 2018 Conference Acceptance Decision**

**Decision:**

Reject

**Comment:**

Pros:
The paper aims to unify classification and novelty detection which is interesting and challenging.

Cons:
- The reviewers find that the work is incremental and contains heuristics. Reviewers find the repurposing of the fake logit in semi-supervised GAN discriminator for assigning novelty strange.
- The experiments presented are weak and authors do not compare with traditional/stronger approaches for novelty detection such as "learning with abstention" models and density models.
GIven the pros and cons, the committe finds the paper to fall short of acceptance in its current form.